# Interfacial Adhesion of a Semi-Interpenetrating Polymer Network-Based Fiber-Reinforced Composite with a High and Low-Gradient Poly(methyl methacrylate) Resin Surface

**DOI:** 10.3390/polym13030352

**Published:** 2021-01-22

**Authors:** Aftab Ahmed Khan, Leila Perea-Lowery, Abdulaziz Abdullah Al-Khureif, Nawaf Abdulrahman AlMufareh, ElZahraa Eldwakhly, Eija Säilynoja, Pekka Kalevi Vallittu

**Affiliations:** 1Dental Biomaterials Research Chair, College of Applied Medical Sciences, King Saud University, Riyadh 10219, Saudi Arabia; aalkhuraif@ksu.edu.sa (A.A.A.-K.); pekval@utu.fi (P.K.V.); 2Department of Biomaterials Science, Institute of Dentistry, University of Turku, FI-20520 Turku, Finland; leila.perea@utu.fi (L.P.-L.); eija.sailynoja@gc.dental (E.S.); 3Department of Pediatric Dentistry & Children with Special Health Care Needs, Ministry of Health, Riyadh 10219, Saudi Arabia; nawaf.almufareh@gmail.com; 4Department of Clinical Dental Science, College of Dentistry, Princess Nourah Bint Abdulrahman University, Riyadh 10219, Saudi Arabia; EAEldwakhly@pnu.edu.sa; 5Department of Fixed Prosthodontics, Faculty of Dentistry, Cairo University, Cairo 12613, Egypt; 6Welfare Division, City of Turku, FI-20520 Turku, Finland

**Keywords:** adhesive interface, fiber-reinforced composite, poly(methyl methacrylate), primer, semi-interpenetrating polymer network, tensile bond strength (TBS)

## Abstract

The research aimed to determine the tensile bond strength (TBS) between polymerized intact and ground fiber-reinforced composite (FRC) surfaces. FRC prepregs (a reinforcing fiber pre-impregnated with a semi-interpenetrating polymer network (semi-IPN) resin system; everStick C&B) were divided into two groups: intact FRCs (with a highly PMMA-enriched surface) and ground FRCs (with a low PMMA gradient). Each FRC group was treated with: StickRESIN and G-Multi PRIMER. These groups were further divided into four subgroups based on the application time of the treatment agents: 0.5, 1, 2, and 5 min. Next, a resin luting cement was applied to the FRC substrates on the top of the photo-polymerized treating agent. Thereafter, weight loss, surface microhardness, and TBS were evaluated. Three-factor analysis of variance (*p* ≤ 0.05) revealed significant differences in the TBS among the FRC groups. The highest TBS was recorded for the intact FRC surface treated with G-Multi PRIMER for 2 min (13.0 ± 1.2 MPa). The monomers and solvents of G-Multi PRIMER showed a time-dependent relationship between treatment time and TBS. They could diffuse into the FRC surface that has a higher PMMA gradient, further resulting in a high TBS between the FRC and resin luting cement.

## 1. Introduction

Earlier dental fiber-reinforced composites (FRCs) comprised reinforced glass fibers incorporated into a dimethacrylate resin. These FRCs showed no challenges related to adhesion when used directly, that is, a new resin-based material was bonded to the FRC via free-radical polymerization with the help of an oxygen-inhibited surface layer. However, in indirect restorations or in those restorations where polymerization is performed at the chair side, such as FRC root canal posts and fixed dental prostheses, the bonding mechanism largely depended on micromechanical locking or adhesion to the exposed glass fiber surfaces that first require chair-side silanization [1,2]. The chair-side and dental-laboratory-made silane-promoted adhesion to glasses and ceramics is prone to hydrolysis with time and cannot guarantee long-term adhesive durability [1,3]. To improve the adhesion, FRCs were subsequently developed by another approach that used semi-interpenetrating polymer network (semi-IPN)-based resin matrix systems.

Semi-IPNs are considered to be a new genre of polymer blends at the macromolecular level, where the polymer chains of linear polymers are cross-linked into a network structure [4]. In semi-IPNs, the possibility of phase separation of the polymers is profoundly reduced because of hindered morphologies of the miscible participating polymers [5]. The inter-network entanglements of the participating polymers are permanent and render the material tough against rupture because of chemical cross-linking [6].

The contemporary dental FRCs comprise continuous unidirectional silanized glass fibers that are impregnated with a resin, which forms a semi-IPN during polymerization [7]. Compared with FRCs having fully cross-linked polymer matrices, the use of semi-IPN-based FRCs can increase the bond strength between the FRC and the resin luting cement along with the veneering resin composite [8]. The improved adhesion between the resin luting cement and the FRC is based on the solubility of the monomer system of the adhesive primer, which allows for the diffusion of the monomers of the resin luting cements or composites into the swelled semi-IPN polymer matrix of the FRC [7,9]. A secure and durable interface between the resin luting cement and the FRC is conducive to the transfer of stress and load from the resin luting cement to the FRC-based prostheses. Additionally, the clinical success of FRC-based prostheses is guaranteed [10,11].

The diffusion of monomers into the linear polymer phase of the poly(methyl methacrylate) (PMMA) of an FRC is governed by the duration of exposure, monomers used in the adhesive resin, primer, resin luting cement, temperature, and polymeric structure of the semi-IPN [12,13,14].

The used adhesive resin and primers are dissolved in various organic solvents, such as acetone, ethanol, tetrahydrofuran, tert-butanol, dimethylsulfoxide, and water [6,15]. However, ethanol is the most commonly used organic hydrophilic solvent that is used either alone or with water as a co-solvent in various commercially available dental adhesives. For a durable bond, it is imperative that the organic solvents of the adhesive primers evaporate from the site of application because the retention of organic solvents may have a deleterious effect on the bonding of the resin-based materials to the substrate [16].

Before polymerization, the resin matrix of the semi-IPN FRC prepreg (a reinforcing fiber pre-impregnated with a resin system) is present in the form of a viscous gel. During the fabrication of the semi-IPN FRC prepreg, the surface layer is enriched with PMMA for effective bonding [6,17]. However, the PMMA gradients of the intact and ground surfaces of FRCs are different [17], which can generally affect the bonding properties of the FRC, the resin luting cement, and resin composites. Therefore, the objective of this study is to determine the tensile bond strength (TBS) between a polymerized semi-IPN FRC with an intact or ground surface and a resin luting cement. Two commercially available surface treatment agents (an adhesive resin and a primer) were used to treat the surfaces of the FRC at different times. The null hypothesis was that TBS is not affected by the intact FRC or ground surface. Moreover, it was also presumed that variables such as the selected adhesive resin or primer and their application time on the polymerized FRC would not affect the bonding properties.

## 2. Materials and Methods

### 2.1. Preparation of FRC Samples

A semi-IPN (based on bisphenol-A-glycidyl methacrylate (*bis*-GMA) and PMMA) FRC prepreg (everStick C&B, Stick Tech Ltd., GC Group, Turku, Finland) with dimensions 4 × 3 × 1 mm^3^ was selected, photo-polymerized, and divided in two categories: intact FRC (surface enriched with high-gradient PMMA) and ground FRC (the glass fibers and inner part of the polymer matrix are exposed). The intact FRC samples were prepared by light curing the FRC while pressing it between the two glass plates. The light curing was performed using a hand-held light polymerization unit (Elipar S10, 3 M ESPE, St. Paul, ML, USA) with a light intensity of 1200 mW/cm^2^ for 40 s by keeping the light curing tip in contact with the glass plate. The ground FRC samples were prepared by the same process, followed by grinding of the flat surfaces to approximately 0.2 mm with a 1200 grit (FEPA) silicon carbide grinding paper to expose the fibers. The substrates were then cleaned in deionized water in an ultrasonic cleaning device (Quantrex 90, L&R Ultrasonics, Kearny, NJ, USA) for 10 min and allowed to set under ambient laboratory conditions (23 °C ± 1 °C) for 60 min.

Next, the intact FRC and ground samples were divided into two groups based on the treatment agent of adhesive resin or primer used: (a) StickRESIN adhesive; and (b) G-Multi PRIMER. Each group was further divided into four subgroups based on the application time of the adhesive resin or primer on the FRC substrate: 0.5, 1, 2, and 5 min protected from light. After the specified application times, the specimens were photo-polymerized again for 40 s in air, leading to the formation of an oxygen-inhibited layer on the surface [18].

### 2.2. Weight Loss Assessment

The weight of the specimens in each group (n = 5, 4 × 3 × 1 mm^3^) was measured to evaluate the weight loss caused by the evaporation of the liquid components of the adhesive resin or primer applied to the FRC substrate. The initial weight of the specimens was measured using a scale (ES 120A, Presica, Dietikon, Switzerland). Then, the adhesive resin or primer was applied to its surface while keeping the specimen on the scale to calculate the change in weight of the adhesive resin and primer. The specimen with the treatment solution was protected from light and kept on the scale for the specified adhesive resin or primer application time. The difference between the initial and final weights was calculated to determine the weight loss caused by the evaporation of the solvents of the monomers during the application time.

### 2.3. Surface Microhardness Test

Surface microhardness testing of the substrate surfaces (n = 5, 4 × 3 × 1 mm^3^) was performed on the surface with the resin luting cement. A 0.3-mm thin layer of resin luting cement was applied over the FRC substrates and was immediately polymerized. By using a Vickers hardness testing machine (Duramin-5; Struers, Westlake, CA, USA), a force of 980.7 mN was applied for 15 s to measure the surface hardness.

### 2.4. Tensile Bond Strength (TBS) Test

The TBS of the FRC–resin luting cement interface was measured using a universal testing machine (Model no. 3369 Instron, Canton, MI, USA) using the specimens dried in air at room temperature. The test was performed at a cross-head speed of 0.5 mm/min using a load cell at 1 kN. For the fabrication of TBS specimen, a mylar sheet was placed over a glass plate and an FRC sample was placed over the mylar sheet. Subsequently, the resin cement (G-CEM LinkAce, GC, Tokyo, Japan) was spread on the mylar sheet on both sides of the FRC (in contact with the FRC), placing the FRC in the middle, like a sandwich, to have resin–FRC–resin. Before light curing the sandwich, another mylar sheet was placed on top of the sandwich. A glass plate was also placed over the mylar sheet, followed by photo polymerization for 40 s. Once the samples were light-cured, the sample blocks were sectioned with a precision diamond saw (Isomet 5000; Buehler Ltd., Lake Bluff, IL, USA) at 1400 revolutions per minute under water cooling to a dimension of 1.0 mm × 1.0 mm × 12.0 mm. A bar-shaped specimen from each group (n = 6) was glued to the grips of a tensile device with cyanoacrylate (Super Glue, Henkel/Loctite, Westlake, CA, USA), and the proprietary software was used to record the failure loads in newtons (N) and the bond strengths in megapascals (MPa).

### 2.5. Statistical Analyses

The effects of variables (adhesive resin or primer, application time, and FRC surface) were analyzed by three-way analysis of variance (ANOVA) and Tukey’s post hoc pairwise comparisons with a significant value of *p* ≤ 0.05. The effect of the adhesive resin or primer’s application time on TBS was further studied using linear regression models. All the statistical calculations were performed using SPSS 23.0 for Windows (SPSS Corporation, Chicago, IL, USA).

## 3. Results

Figure 1 shows the weight loss (in %) caused by the evaporation of solvents and monomers from the adhesive resin and primer applied to the FRC substrate for different periods. The G-Multi Primer applied to the ground FRC surface for 5 min showed the highest weight loss (85.95%), whereas the StickRESIN adhesive showed little or no susceptibility to evaporation. Hence, most of the FRC substrates treated with StickRESIN for different times showed no (0%) weight loss within the limits of accuracy of the method.

Table 1 presents the Vickers microhardness (VHN) values with mean and standard deviations of the intact FRC and ground FRC surfaces treated with the adhesive resin or primer. A marked difference in Vickers hardness was observed for the resin luting cement that was applied on the top of the treated intact or ground FRC surfaces. The highest Vickers hardness was recorded for the ground FRC surface treated with G-Multi PRIMER for 30 s, whereas the lowest one was recorded for the intact FRC surface treated with StickRESIN for 2 min. Overall, the hardness increased as the treatment time of the FRC substrate by the adhesive resin or primer increased, irrespective of the adhesive primer used (Figure 2).

Table 2 summarizes the mean and standard deviation of TBS values (in MPa) of the study groups. Among all the study groups, the intact FRC treated with G-Multi PRIMER for 2 min shows the highest TBS (13.0 MPa ± 1.2 MPa), whereas the ground FRC treated with G-Multi PRIMER for 30 s shows the lowest mean TBS (3.5 MPa ± 2.0 MPa). Furthermore, three-factor ANOVA revealed significant differences in the TBS of the different groups: FRC surfaces (intact or ground) (*p* < 0.001), primers used (*p* = 0.047), and treatment time (*p =* 0.010). The interactive effect of the variables, such as the FRC surface and adhesive primer, had a significant effect on TBS (*p* < 0.001). However, the interactive effect of the FRC surface and the treatment time; the adhesive resin or primer and the treatment time; and the joint effect of the FRC surface, the adhesive resin or primer, and the treatment time had an insignificant effect on TBS (*p* > 0.05).

Table 3 shows the regression analysis in the split mode between the treatment time and TBS of the FRC surfaces with a resin luting cement layer. The correlation coefficients (R) of the G-Multi PRIMER for the intact FRC surface (R = 0.657) and the ground FRC surface (R = 0.731) are high. The linear relationships suggest that both the G-Multi PRIMER-treated intact and ground FRC surfaces affected the treatment time, which led to an increase in TBS. However, when StickRESIN was applied on the intact FRC or ground FRC surface, no correlation was found (R = 0.197 and R = 0.118, respectively) between the two variables.

The regression coefficient table provides more necessary information to predict the variation in TBS with treatment time (Table 4). The data reveal that G-Multi PRIMER significantly affected the TBS of both the intact and ground FRC specimens (*p* = 0.002 and *p* < 0.001, respectively). However, the treatment time has an insignificant effect on both intact FRC and ground FRC specimens treated using StickRESIN (*p* > 0.05).

## 4. Discussion

This laboratory study analyzed the interfacial TBS between a semi-IPN-based FRC and a resin luting cement. The working hypothesis was contradicted because the adhesive resin and primer with different monomer systems and solvents exhibited a significant effect on TBS between the intact FRC along with ground substrates and the resin luting cement.

The elimination of solvents before light curing is crucial because residual solvents may deteriorate and interfere with the polymerization process at the adhesive interface [15]. Usually, acetone, ethanol, and water are the main solvents in commercial formulations [19]. We observed that G-Multi PRIMER tended to evaporate at room temperature, which possibly occurs because of the high content of ethyl alcohol used in the formulation (up to 90%). Although the evaporation of solvent from G-Multi PRIMER was minimal at 0.5 min (5.08%), a weight loss of 85.95% was observed at 5 min because of solvent evaporation. This phenomenon suggests that the evaporation of ethanol at room temperature (~23 °C ± 1 °C) is slow; however, the vapor pressure of ethanol at 20 °C is 5.95 kPa. However, this pressure increases to 53.3 kPa at 63.5 °C [20]. Thus, at a higher room temperature, the evaporation might have been quicker because of an increase in the vapor pressure of ethanol and we would have obtained good TBS values for the treatment time groups of 0.5 min and 1 min. In contrast, the StickRESIN adhesive tended to infiltrate the FRC substrate. This process occurs possibly because of the formulation of StickRESIN, which contains only *bis*-GMA and triethylene glycol dimethacrylate (TEGDMA) monomers with an initiator system and does not contain any solvent with a considerably low vapor pressure.

The surface microhardness test is a relatively simple and effective technique for determining some basic mechanical features of a material [13,14]. The hardness data suggest that the ground FRC surfaces are harder than the intact FRC surfaces, irrespective of the treatment agent used. This phenomenon can be attributed to the characteristic property of linear polymers, that is, ductility. PMMA is a ductile resin polymer [15] with a characteristic low hardness, and its polymer chains are bound together predominantly by weak chemical bonds such as van der Waals forces [21]. In contrast, the polymer chains of the cross-linked *bis*-GMA–TEGDMA copolymer are held by covalent bonds in the semi-IPN system. Because the PMMA gradient is higher in intact FRC than in ground FRC surfaces, the hardness values of the intact FRC groups were lower than those of the ground FRC groups.

In the semi-IPN-based FRC prepregs, such as everStick C&B, there is a variation in the ratio of the cross-linked dimethacrylate component to the PMMA component. The PMMA gradient is higher at the intact surface than at the ground surface [6,17]. We observed that the application of G-Multi PRIMER on the intact FRC substrate enhanced the TBS of the FRC to the resin luting cement. This phenomenon might be attributed to the presence of an enriched PMMA layer. The large amount of ethanol in G-Multi PRIMER swells the PMMA, as the solubility parameter of ethanol (12.92 (cal/cm^3^)^1/2^) is very close to that of PMMA (8.9–12.7 (cal/cm^3^)^1/2^) [6]. This occurrence may facilitate the diffusion of the phosphate ester monomers of the primer and the dimethacrylate component of the resin luting cement into the PMMA layer of the FRC. The polymer chain mobility and radical diffusion rates have already been determined using ethanol [22], and the effect of ethanol on PMMA has been thoroughly investigated in previous studies [23,24]. In contrast, the adhesion of the resin luting cement to the ground FRC was inadequate with the G-Multi PRIMER treatment. This phenomenon can possibly be attributed to the fact that the intact surface comprises 100% polymeric material, whereas the ground surface comprises 50% polymer and 50% exposed silicate glass, that is, fiber [17]. This process suggests that a lack of the PMMA component in ground FRC leads to insufficient interaction with the phosphate ester monomer and the dimethacrylate component of the resin luting cement.

We speculate that the StickRESIN adhesive resin could not adequately dissolve the PMMA-enriched intact FRC surface. According to the material’s data sheet, StickRESIN contains *bis*-GMA and has a molecular weight of 512 g/mol [4]. This attribute might render the adhesive resin viscous and reduce the mobility of *bis*-GMA during polymerization [22,25]. Additionally, the photoinitiator system and the subsequent spontaneous initiation of polymerization of StickRESIN might have prevented the dissolution by *bis*-GMA [8,9], thereby lowering the TBS of the adhesive interface for this group, irrespective of the time for which the adhesive resin was applied to the intact FRC substrate. This result is in line with the previously published results [15].

However, the TBS of the ground FRC surface treated with StickRESIN to the resin luting cement was higher than that of the ground FRC surface treated with G-Multi PRIMER. The data suggest that the presence of a highly cross-linked resin component in the ground FRC is not conducive to the formation of a strong bond between the resin luting cement and ground FRC. The dimethacrylate resin component of the polymer matrix of the FRC polymerizes to a highly cross-linked component [26], and its high cross-linking density results in the insufficient adhesion of the resin luting cement to the FRC. In the semi-IPN structure with PMMA, the presence of PMMA enables the dissolution of the PMMA surface [27,28]. It is worth noting that the bond strength values in this study were obtained by testing the tensile strength, not by the so-called shear bond strength test. Therefore, the results in MPa cannot be compared to the majority of the reported bond strength tests. TBS testing has been reported to be more appropriate for studying the interfaces of dental materials [29,30]. The so-called shear bond strength values in MPa are generally higher than the corresponding TBS values.

The linear regression model in the split mode clearly supports the effect of application time of G-Multi PRIMER on both intact FRC (R = 0.657) and ground FRC (R = 0.731) specimens, hence showing a linear relation between the priming time and TBS. The non-linear correlation between the treatment time and TBS in the case of StickRESIN adhesive resin can reflect the inefficiency of dissolution by this type of adhesive resin on the FRC surfaces.

In future, it would be interesting to evaluate the effect of temperature or other methods to enhance the efficacy of adhesive resins and primers during their application on FRC surfaces. This study evaluated only one type of resin luting cement. However, in future laboratory studies, the use of different resin cement systems would be exciting and promising.

## 5. Conclusions

The following conclusions were drawn from the findings of this laboratory study:Prolonging the treatment time of the semi-IPN FRC by adhesive resin or primer enhanced the TBS between the resin luting cement and the FRC, irrespective of the treatment agent used.The highest TBS between the intact FRC and the resin luting cement was obtained with the G-Multi PRIMER treatment with 2 min of evaporation of the solvent of the primer.

## Figures and Tables

**Figure 1 polymers-13-00352-f001:**
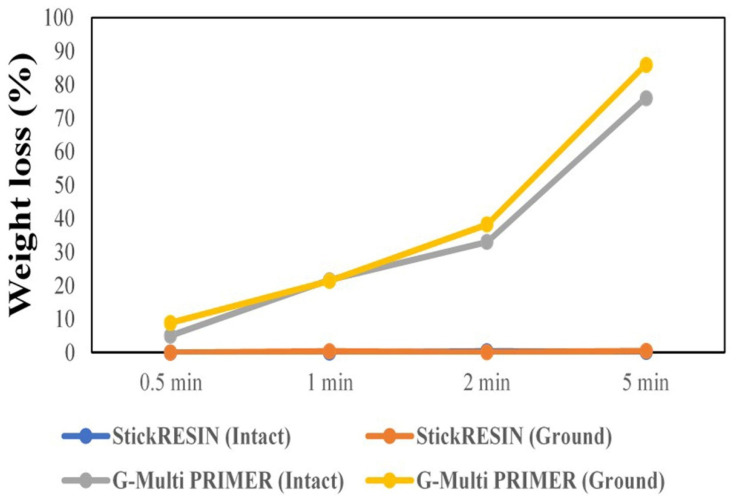
Weight loss of the fiber-reinforced composite (FRC) (in %) as a function of the evaporation of monomers and solvents at different time points.

**Figure 2 polymers-13-00352-f002:**
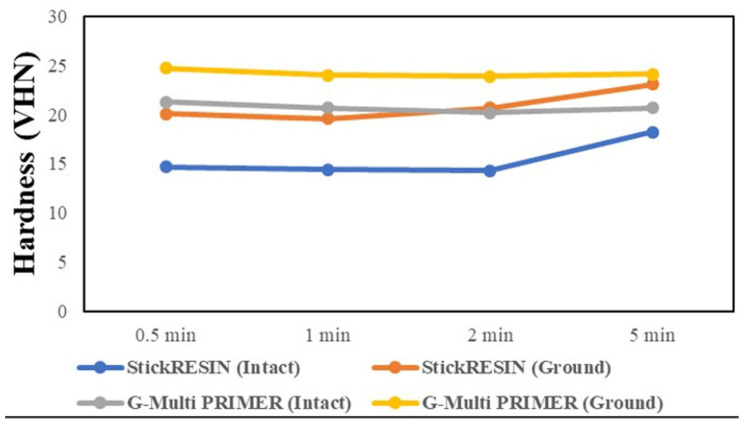
Hardness of the FRC substrate surface with the resin luting cement as a function of adhesive resin or primer application time.

**Table 1 polymers-13-00352-t001:** Surface microhardness data of intact FRC and ground FRC specimens with different treatment agents (G-Multi PRIMER or StickRESIN) at different treatment times.

Treatment Time	Vickers Hardness Number (VHN)			
	G-Multi PRIMER (Intact)	G-Multi PRIMER (Ground)	StickRESIN (Intact)	StickRESIN (Ground)
30 s	21.3 ± 1.1 ^A,B^	24.6 ± 1.4 ^A,C,D^	14.7 ± 0.5 ^B,C,E^_a_	20.1 ± 1.2 ^D,E^_d_
1 min	20.7 ± 0.8 ^F,G,H^	24.0 ± 0.4 ^F,I,J^	14.4 ± 0.3 ^G,I,K^_b_	19.6 ± 0.8 ^H,J,K^_e_
2 min	20.2 ± 1.4 ^L,M^	23.9 ± 0.3 ^L,N,O^	14.3 ± 0.3 ^M,N,P^_c_	20.7 ± 0.6 ^O,P^_f_
5 min	20.6 ± 1.7 ^Q,R,S^	24.1 ± 1.5 ^Q,T^	18.1 ± 0.5 ^R,T,U^_a,b,c_	23.1 ± 0.5 ^S,U^_d,e,f_

Same superscript uppercase letters demonstrate the significant difference between the FRC surfaces and priming groups (*p* ≤ 0.05). Moreover, same subscript lowercase letters demonstrate the significant difference between the different treatment time groups (*p* ≤ 0.05).

**Table 2 polymers-13-00352-t002:** Tensile bond strength (TBS) data (with descriptive and inferential statistics) of the study groups wherein the resin luting cement was applied to the FRC surface with different treatment agents (G-Multi PRIMER or StickRESIN) at different treatment times.

Treatment Time	Tensile Bond Strength (MPa)			
	G-Multi PRIMER (Intact)	G-Multi PRIMER (Ground)	StickRESIN (Intact)	StickRESIN (Ground)
30 s	9.0 ± 2.0 ^A^_a,b_	3.5 ± 2.0 ^A^^,B,C^_d,e,f_	9.5 ± 1.1 ^B^	8.4 ± 2.0 ^C^
1 min	9.5 ± 1.9 ^D^_c_	6.0 ± 1.0 ^D^_d_	9.3 ± 1.5	8.4 ± 2.8
2 min	13.0 ± 1.2 ^E^_a,c_	5.9 ± 0.9 ^E,F,G^_e_	10.1 ± 3.1 ^F^	9.9 ± 2.6 ^G^
5 min	12.3 ± 1.7 _b_	7.4 ± 0.5 _f_	10.9 ± 4.9	9.0 ± 4.8

See Table 1.

**Table 3 polymers-13-00352-t003:** Linear regression model in the split mode according to the treating conditions to determine the effect of treatment time on intact FRC and ground FRC surfaces.

Study Groups	Model	R	R-Square	Adjusted R-Square	Std. Error of the Estimate	Change Statistics			
						R-Square Change	F Change	df1	df2
G-Multi PRIMER (intact)	1	0.657 ^a^	0.431	0.400	1.84373	0.431	13.659	1	18
G-Multi PRIMER (ground)	1	0.731 ^a^	0.534	0.508	1.30039	0.534	20.648	1	18
StickRESIN (intact)	1	0.197 ^a^	0.039	−0.015	2.90579	0.039	0.727	1	18
StickRESIN (ground)	1	0.118 ^a^	0.014	−0.041	3.10458	0.014	0.256	1	18

^a^ Predictors: Treatment time.

**Table 4 polymers-13-00352-t004:** Regression coefficient table in the split mode showing positive and negative correlations between each independent variable against treatment time.

Study Groups		Unstandardized Coefficients		Standardized Coefficients	t	Sig.
		B	Std. Error	Beta		
G-Multi PRIMER (intact)	(Constant)	7.554	1.010	°	7.480	0.000
	Treatment time	1.363	0.369	0.657	3.696	0.002
G-Multi PRIMER (ground)	(Constant)	2.753	0.712	°	3.865	0.001
	Treatment time	1.182	0.260	0.731	4.544	0.000
Stick RESIN (intact)	(Constant)	8.720	1.592	°	5.479	0.000
	Treatment time	0.495	0.581	0.197	0.852	0.405
Stick RESIN (ground)	(Constant)	8.161	1.700	°	4.799	0.000
	Treatment time	0.314	0.621	0.118	0.506	0.619

Dependent Variable: TBS.

## Data Availability

The data presented in this study are available on request from the corresponding author.

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
