# Peer review of "Interfacial Adhesion of a Semi-Interpenetrating Polymer Network-Based Fiber-Reinforced Composite with a High and Low-Gradient Poly(methyl methacrylate) Resin Surface"

_polymers, 2021, doi:10.3390/polym13030352_

Round 1

Reviewer 1 Report

The manuscript “Interfacial adhesion of semi-interpenetrating polymer network- 2 based fiber-reinforced composite with high and low-gradient 3 poly(methyl methacrylate) resin surface” present new significant information about the new Adhesive interface.

Before acceptance, some issued need to be corrected and explained:

  1. The figure 1 and 2 required insertion of error mean data
  2. I recommended delate “In future…” paragraph from the end of discussion. If you know what other text need to be done – why you did not provide these experiment in current research
  3. The Conclusion section should be rewrite due to complexity
  4. The “CLINICAL IMPLICATIONS” section is very important but you did not discuss any clinical relevance in the text. Please, provide some discussion using existing data.

Author Response

The authors are thankful to reviewer for his time and effort in reviewing the paper. Our responses are as under:

  1. By inserting the error means, the present data is messing up. We suppose, there is no need to show error mean as our intention was just to show the trend in the graph, not the statistical difference between the groups.
  2. We, the authors suppose that "future suggestion" is an important aspect of a lab based study for continuing the same kind of work. Future suggestions help other investigators in proceeding the work further. All the parameters can not always be covered in a single study.
  3. The conclusion section is very much clear, concise and meaningful.
  4. Yes, we agree with the reviewer. We have now thoroughly modified the "clinical implications" section.

Reviewer 2 Report

  1. The abstract is not structured correctly. It should contain the same parts as the article, but in abbreviated form. To be edited!
  2. CLINICAL IMPLICATIONS (line 40) are not in appropriate place! It can be added to the results, discussion or conclusion!
  3. The introduction is well described, but the authors from the discussion part (from 18 to the end) should be added to it!
  4. The materials and methodology are well described and arranged!
  5. The results are presented and illustrated very well!
  6. The authors of the discussion should be added to the introduction! The correct structure requires acquainting the reader with the work of the respective researcher in the introduction, and then comparing the results in the discussion.
  7. The discussion is voluminous enough, with a large number of authors.
  8. The conclusion is short and concise, without unnecessary information.
  9. The city and country can be added to ACKNOWLEDGEMENT.
  10. The authors in reference are a sufficient number (before 2010 - 16, after 2010 - 15).
  11. Self-citation - 14 articles (6, 8, 9, 10, 12, 14, 15, 17, 23, 24, 25, 27, 28, 29). This shows that the authors are seriously and for many years dealing with the problem and I think they can remain as confirmatory material for the article.

Author Response

The authors are thankful to reviewer for his/her valuable suggestions to improve the paper further. Our responses against each comment from the reviewer are as under:

  1. We have now structured the abstract correctly.
  2. We have seem in previous published papers in the same journal that "Clinical Implications" section is separately present under the Abstract. Hence, we followed the same formatting style.
  3. Now the papers (3 or 4 papers) from the discussion part are included in the Introduction part.
  4. Thank you.
  5. Thank you. 
  6. Now done according to the reviewer's suggestion.
  7. Yes, Discussion is lengthy because quite the number of testing parameters were compared and contrasted with the previous studies.
  8. Thank you once again.
  9. We have to strictly follow the acknowledgement statement as received from the funding body. We can not add the city and country in it.
  10. Yes, true! The authors' have numerous related published papers in peer reviewed journals.
  11. Thank you once again.

Round 2

Reviewer 1 Report

Dear Authors!

Thank you for your reply. After analysis of revised version I suggest it should be suitable for publication